# Biological and Genetic Characterizations of a Novel Lytic ΦFifi106 against Indigenous *Erwinia amylovora* and Evaluation of the Control of Fire Blight in Apple Plants

**DOI:** 10.3390/biology12081060

**Published:** 2023-07-28

**Authors:** Jaein Choe, Byeori Kim, Mi-Kyung Park, Eunjung Roh

**Affiliations:** 1School of Food Science and Biotechnology, Food and Bio-Industry Research Institute, Kyungpook National University, Daegu 41566, Republic of Korea; jane1226@knu.ac.kr; 2Crop Protection Division, National Institute of Agricultural Sciences, Rural Development Administration, Wanju 55365, Republic of Korea; byeori4689@naver.com

**Keywords:** lytic phage, *Erwinia amylovora*, fire blight, apple plant, disease incidence, disease severity

## Abstract

**Simple Summary:**

The devastating phytobacterium, *Erwinia amylovora,* causes fire blight in the *Rosaceae* family. In this study, a novel ΦFifi106 was characterized and its efficacy was evaluated for the control of fire blight. ΦFifi106 showed a specific infectivity to indigenous *E. amylovora* and *E. pyrifoliae*. ΦFifi106 exhibited an excellent lytic activity under broad temperatures, pHs, and exposure to ultraviolet irradiation. Finally, the pretreatment of ΦFifi106 efficiently reduced both the development and the disease severity of fire blight in apple plants compared to those of commercial products, AgriPhage-FireBlight and Bramycin. Our findings have proven that ΦFifi106 was a potential alternative to control fire blight in apple plants.

**Abstract:**

*Erwinia amylovora* is a devastating phytobacterium causing fire blight in the *Rosaceae* family. In this study, ΦFifi106, isolated from pear orchard soil, was further purified and characterized, and its efficacy for the control of fire blight in apple plants was evaluated. Its genomic analysis revealed that it consisted of 84,405 bp and forty-six functional ORFs, without any genes encoding antibiotic resistance, virulence, and lysogenicity. The phage was classified into the genus *Kolesnikvirus* of the subfamily *Ounavirinae*. ΦFifi106 specifically infected indigenous *E. amylovora* and *E. pyrifoliae*. The lytic activity of ΦFifi106 was stable under temperature and pH ranges of 4–50 °C and 4–10, as well as the exposure to ultraviolet irradiation for 6 h. ΦFifi106 had a latent period of 20 min and a burst size of 310 ± 30 PFU/infected cell. ΦFifi106 efficiently inhibited *E. amylovora* YKB 14808 at a multiplicity of infection (MOI) of 0.1 for 16 h. Finally, the pretreatment of ΦFifi106 at an MOI of 1000 efficiently reduced disease incidence to 37.0% and disease severity to 0.4 in M9 apple plants. This study addressed the use of ΦFifi106 as a novel, safe, efficient, and effective alternative to control fire blight in apple plants.

## 1. Introduction

Over 200 species of phytobacteria cause significant crop losses during harvesting, transportation, and storage [1]. The major ones include those from the genera *Agrobacterium*, *Dickeya*, *Erwinia*, *Pseudomonas*, *Pectobacterium*, *Ralstonia*, *Xanthomonas*, and *Xylella* [1,2]. Since *Erwinia amylovora* was identified as the first phytobacterium in the late 1800s, it has been reported to cause fire blight in a wide range of species belonging to the *Rosaceae* family, such as apple, pear, blackberry, and raspberry [3]. Owing to its rapid dissemination across the world in the last two centuries [3,4], *E. amylovora* has been categorized as a quarantine organism in several Asian and European countries [1].

*E. amylovora* is a Gram-negative, motile, and rod-shaped bacterium belonging to the family *Enterobacteriaceae*. Generally, two factors including exopolysaccharides amylovoran and the Hrp type III secretion system are involved in expressing its virulence and systemic infection in *E. amylovora* [4]. Although there is no clear explanation of invasion mode, *E. amylovora* moves rapidly through the intercellular spaces of the parenchyma within a target plant until reaching the xylem vessels. It provokes extensive and systematic infection with lesions, wilting, ooze secretion, and death of blossoms, twigs, and fruits in the entire plant, resulting in color change to dark green, brown, or black [3,5]. Once the fire blight has occurred, the best treatment is to eradicate the diseased plants. Alternatively, preventive treatments are recommended, such as copper-based pesticides (copper hydroxide, copper sulfate basic, and oxine-copper) and antibiotics (streptomycin, kasugamycin, and oxytetracycline) [6,7]. Excluding their effectiveness in terms of efficacy and cost, these persistent treatments can lead to bacterial resistance as well as the accumulation of these chemicals in the environment and humans [8,9,10]. These limitations have prompted to search for a novel, safe, and eco-friendly alternatives for controlling *E*. *amylovora* [11].

Bacteriophages (phages) are viruses that infect only target bacteria without affecting any coexisting bacteria and are ubiquitous in nature (10^31^ phages) [12,13]. Particularly, lytic phages can replicate and assemble their parts using the host’s system after attaching and injecting their DNA into the host. Finally, the assembled phages in the host are released via lysis with an excellent burst size, which is contradictory to the destination of lysogenic phages existing as prophages within the host [12]. Thus, lytic phages are considered novel, efficient, and eco-friendly alternatives to substitute for chemical pesticides and antibiotics [8,9]. Despite numerous efforts for several decades, only two commercial phage products, AgriPhage-Fire Blight (Omnilytics Inc., Salt Lake City, UT, USA) and Erwiphage PLUS (Environvest Ltd., Kertváros, Hungary), are currently available for controlling *E. amylovora* [8].

Nonetheless, the majority of studies on *E. amylovora*–specific phages have focused on their isolation and characterization [7,10,14,15]. Only a few studies have addressed the possibility of practical application by demonstrating phages’ in vivo efficacies in controlling the development of fire blight using apple plants [16], apple flowers [17], and pear slices [11,18,19]. In our previous study [7], nine *E*. *amylovora*–specific phages were isolated from apple and pear orchards in Korea. Among them, ΦFifi106 isolated from pear soil was further purified using an indigenous *E. amylovora* YKB 14808 due to the excellency of its lytic activity. The biological and genomic characteristics of the purified phage were investigated. Finally, its biocontrol efficacy was objectively evaluated with those of other commercial products using three-month-old M9 apple (*Malus* spp.) plants.

## 2. Materials and Methods

### 2.1. Bacterial Strain and Culture Condition

*E. amylovora* YKB 14808 was used as a host strain in this study. Thirty-one phytobacteria (Table 1) were used for host range analysis, including ten *E. amylovora*, eleven *E. pyrifoliae,* six *Pectobacterium carotovorum*, three *Xanthomonas arboricola*, and one *Xanthomonas campestris*. Each phytobacterium was cultivated in tryptic soy broth (TSB; Difco Laboratories Inc., Sparks, MD, USA) at 28 °C for 16 h. After centrifugation at 6000× *g* at 4 °C for 4 min, the pellet was washed thrice using phosphate-buffered saline (pH 7.4; Life Technologies Ltd., Paisley, UK). The final concentration of each bacterial culture was determined using a standard curve constructed by measuring the optical density at 600 nm.

### 2.2. Propagation and Purification of ΦFifi106

For the propagation of ΦFifi106, 3 mL of TSB containing 1% (*v*/*v*) host culture was incubated at 28 °C for 2.5 h. Afterwards, 100 μL of ΦFifi106 [10^8^ plaque-forming unit (PFU)/mL] was added and further incubated in the same conditions. Following centrifugation at 6000× *g* at 4 °C for 20 min, the supernatant was filtered through a 0.20-μm cellulose acetate filter (Advantec Toyo Kaisha Ltd., Tokyo, Japan). The filtrate was placed in 8 mL of TA broth (8 g/L nutrient broth, 5 g/L NaCl, 0.2 g/L MgSO_4_, 0.05 g/L MnSO_4_, and 0.15 g/L CaCl_2_) containing 1% host culture and incubated prior to centrifugation at 6000× *g* at 4 °C for 20 min. All procedures were repeated by increasing the amount of TA broth. Finally, the propagated filtrate was precipitated by mixing with 10% (*w*/*v*) polyethylene glycol 8000 (Sigma-Aldrich Co., Ltd., St. Louis, MO, USA) and 10 mL of 1 M NaCl at 4 °C for 16 h. Following centrifugation at 7200× *g* at 4 °C for 20 min, the pellet was suspended in sodium chloride–magnesium sulfate (SM) buffer (50 mM/L Tris-HCl, 100 mM NaCl, and 10 mM MgSO_4_; pH 7.5) and purified through CsCl gradient ultracentrifugation at 39,000× *g* at 4 °C for 2 h. After dialysis with SM buffer at 4 °C for 6 h, the final concentration of the purified ΦFifi106 was determined by pouring a mixture of 100 μL of phage and 4 mL of TA soft agar (4 g/L agar, 8 g/L nutrient broth, 5 g/L NaCl, 0.2 g/L MgSO_4_, 0.05 g/L MnSO_4_, and 0.15 g/L CaCl_2_) containing 5% (*v*/*v*) host culture onto a tryptic soy agar (TSA; Difco Laboratories Inc., Sparks, MD, USA) plate, followed by incubation at 28 °C for 16 h (plaque assay). The purified ΦFifi106 was stored at 4 °C prior to use.

### 2.3. Morphological Analysis of ΦFifi106

ΦFifi106 was adsorbed on a carbon-coated copper grid and negatively stained with 2% (*v*/*v*) aqueous uranyl acetate (Sigma-Aldrich Co., Ltd.). The morphological characteristics of ΦFifi106 were observed and analyzed using transmission electron microscopy (TEM; LEO 912AB, LEO Electron Microscopy Inc., Thornwood, NY, USA) at an accelerating voltage of 100 kV with 160,000× magnification.

### 2.4. Genomic Sequencing and Annotation of ΦFifi106

The genomic DNA of ΦFifi106 was extracted and purified using a Phage DNA Isolation Kit (Norgen Biotek Co., Ltd., Thorold, ON, Canada) according to the manufacturer’s instructions. The purified genomic DNA was sequenced using the HiSeq X Ten platform (Illumina Inc., San Diego, CA, USA) and trimmed using Trimmomatic [20] to remove adapter sequences and low-quality reads. The qualified sequences were de novo assembled using a SPAdes genome assembler (Illumina Inc.) [21]. The open reading frames (ORFs) were predicted and annotated using the prokaryotic genome annotation pipeline (Prokka) [22] and BLASTP. The complete genome sequence of ΦFifi106 was deposited at the GenBank database under accession number OR284297. The presences of genes encoding tRNA, antimicrobial resistance, and virulence were analyzed using tRNAscan-SE (v2.0) [23], ResFinder (v4.1), and VirulenceFinder (v2.0) [24], respectively. The PhageAI tool (v0.10.0) was used to predict the lytic and lysogenic cycles [25]. A genome map of ΦFifi106 was generated using the Blast Ring Image Generator [26].

### 2.5. Phylogenetic and Comparative Genomic Analyses of ΦFifi106

Phylogenetic analysis of ΦFifi106 was performed using the Virus Classification and Tree Building Online Resource [27]. The average nucleotide identity (ANI) values of ΦFifi106 with forty-four *E. amylovora*–specific myophages were determined using the OrthoANI tool (v0.93.1) [28]. Heat map analysis was conducted based on the ANI values using the GraphPad Prism (GraphPad Software Inc., San Diego, CA, USA). Finally, the sequence similarity between ΦFifi106 and two International Committee on Taxonomy of Viruses (ICTV)-classified *Kolesnikvirus* phages was analyzed using tBLASTx with default settings in Easyfig software (v2.2.5).

### 2.6. Host Range Analysis of ΦFifi106

The host range of ΦFifi106 was investigated via dot assay using phytobacteria (Table 1). Briefly, 200 μL of each bacterial culture was mixed with 4 mL of TA soft agar and overlaid on a TSA plate. Then, 10 μL of diluted ΦFifi106 (10^7^ PFU/mL) was dotted on the plate and incubated at 28 °C for 16 h. The formation of a clear zone indicated the susceptibility of the phage to the tested phytobacterium.

### 2.7. Temperature, pH, and Ultraviolet Irradiation Stabilities of ΦFifi106

ΦFifi106 (10^7^ PFU/mL) was suspended in 1 mL SM buffer to investigate the effects of temperature, pH, and ultraviolet (UV) irradiation. For temperature effect, the ΦFifi106 suspension was separately incubated at 4, 10, 20, 30, 40, 50, and 60 °C for 1 h. To measure the pH effect, ΦFifi106 was suspended in 1 mL of SM buffer at pHs of 4, 5, 6, 7, 8, 9, 10, and 11 and incubated at 22 °C for 1 h. For the UV effect, the ΦFifi106 suspension was placed 30 cm from the UV light source and exposed to UV irradiation (λ = 250−1700 nm; 100 mW/cm^2^) for 1, 2, 3, 4, 5, and 6 h at 22 °C using a solar simulator (Newport Co., Ltd., Irvine, CA, USA). Finally, the plaque assay was performed to compare the phage concentration.

### 2.8. One-Step Growth Curve Analysis of ΦFifi106

To determine latent period and burst size of ΦFifi106, 50 mL of the TSB containing 1% (*v*/*v*) host culture was incubated at 28 °C until it reached ~10^8^ colony-forming units (CFU)/mL. Following the addition of ΦFifi106 suspension at a multiplicity of infection (MOI) rate of 0.01, the mixture was incubated at 28 °C for 20 min to allow adsorption of ΦFifi106 to the host and then centrifuged at 11,400× *g* for 10 min to remove the unabsorbed phages. The infected bacterial pellet was resuspended in 50 mL of TSB prior to incubation at 28 °C for 1 h. At every 5 min interval, 1 mL of the incubated mixture was collected and centrifuged at 9700× *g* for 1 min followed by filtration. Afterwards, the plaque assay was performed to determine the latent period and burst size of the phage. The latent period was defined as the time interval between the infection and the release of the first phage from the host. The burst size was calculated as the ratio of the final count of free phage particles after the rise phase to their initial count during the latent period [29].

### 2.9. Time Killing Assay of ΦFifi106 with Different MOIs against E. amylovora

The lytic activity of ΦFifi106 was compared with different MOIs. After sub-culturing the host suspension at 28 °C for 2 h, 180 μL of the diluted host suspension (10^5^ CFU/mL) was added to each well of a 96-well plate. ΦFifi106 was placed with different MOIs of 0.001, 0.1, 10, and 1000, and the host growth was monitored at 1 h intervals for 16 h using a microplate reader (Multiskan FC Microplate Photometer; Thermo Fisher Scientific Inc., Waltham, MA, USA).

### 2.10. In Vivo Evaluation of ΦFifi106 Efficacy against Fire Blight Development in M9 Apple Plants

Three-month-old M9 apple (*Malus* spp.) plants in a tissue culture vessel (72 × 72 × 100 mm, SPL Life Sciences Co., Ltd., Pocheon, Republic of Korea) filled with Murashige and Skoog medium supplemented with 0.8% (*w*/*v*) agar and 3% (*w*/*v*) sucrose was provided by the Korea Agriculture Technology Promotion Agency (Iksan, Republic of Korea). For the comparison of the efficacy, AgriPhage-FireBlight (Kyungnong Co., Ltd., Seoul, Republic of Korea) and Bramycin (Farm Hannong Co., Ltd., Seoul, Republic of Korea) were used for the comparison of their efficacies with ΦFifi106, and sterilized water was used as a control. One milliliter each of ΦFifi106 (10^8^ PFU/mL), AgriPhage-FireBlight (10^10^ PFU/mL), Bramycin (100 ppm), and sterilized water was separately sprayed on the surface of individual M9 apple plants using a compression spray. After 2 h of drying, 1 mL of *E. amylovora* suspension (10^5^ CFU/mL) was sprayed on the surface of each M9 apple plant and kept at 25 ± 5 °C for 2 weeks under a relative humidity of ~85%. The appearance and development of fire blight in its leaves and stems were observed and quantified for 2 weeks. The incidence was calculated by counting the occurrence of symptomatic fire blight in the leaves and stems from the total number of samples observed in each apple plant, in accordance with the protocol established by the Commonwealth Mycological Institute [30]. The disease severity index was determined using a scale of 0 to 4 based on the visual appearance of the area of damage in leaves and stems, where 0 = healthy tissue, 1 = 1–25% of leaves and stems affected, 2 = 26–50% of leaves and stems affected, 3 = 51–75% of leaves and stems affected, and 4 = 76–100% of leaves and stems affected [31]. All experiments were performed in triplicate with nine individual M9 apple plants per group.

### 2.11. Statistical Analysis

All data were expressed as the mean ± standard deviation after triplicate experiments. GraphPad and InStat V.3 (GraphPad Software Inc.) were used to perform the statistical analysis. The means were compared using one-way analysis of variation for more than two groups and Student’s paired *t*-test for comparison between two groups. A *p*-value of less than 0.05 or 0.001 was considered statistically significant.

## 3. Results

### 3.1. Propagation, Purification, and Morphological Characterization of ΦFifi106

ΦFifi106 was propagated and purified at a final concentration of (1.7 ± 0.3) × 10^11^ PFU/mL. From the TEM image (Figure 1), the phage possessed an icosahedral head with length and width of 79.8 ± 4.3 nm and 74.2 ± 2.1 nm, respectively, and a tail length of 114.1 ± 5.2 nm. The purified phage belonged to a myophage group within the class *Caudoviricetes*, as it possessed a rigid and contractile tail by comparison with other reported studies on myophages [10,32,33] and siphophages [34,35].

### 3.2. Genomic Characterization of ΦFifi106

The genome of ΦFifi106 consisted of 84,405 bp with GC content of 43.4%. In total, sixty-eight hypothetical ORFs, forty-six functional ORFs, and twenty-six tRNAs were predicted in the phage genome (Figure 2). The functional ORFs of the phage were categorized into four groups encoding twenty structure-related proteins, four host lysis-related proteins, eighteen nucleotide metabolism-related proteins, and four unclassified functional proteins (Figure 2). The further annotation analysis based on the ResFinder, VirulenceFinder, and PhageAI databases revealed that ΦFifi106 did not contain any genes encoding antibiotic resistance, virulence, or lysogenicity, respectively.

### 3.3. Phylogenetic and Genomic Comparative Analysis of ΦFifi106

From the comparison of ΦFifi106 with forty-four *E*. *amylovora*–specific myophages, the phage was clustered in the same branch with six phages (indicated by an aqua box), including vB_Eam_M-M7, Hena2, phiEa104, phiEa21-4, SunLiRen, and Roscha 1 within the genus *Kolesnikvirus* of the subfamily *Ounavirinae* (Figure 3). Heat map analysis based on the ANI values (Figure 4) confirmed the homologous relationship between ΦFifi106 and the aforementioned six phages, with specific similarities to vB_EamM-M7 (98.8%), Hena2 (98.1%), phiEa104 (87.7%), phiEa21-4 (87.4%), SunLiRen (87.3%), and Roscha 1 (88.0%). However, ΦFifi106 did not have any homologous relationship with the other thirty-seven *E. amylovora*–specific myophages, except for vB_EamM_Asesino (59.7%). Finally, genomic comparative analysis (Figure 5) confirmed genetic differences in the annotated proteins with vB_EamM-M7 (the most similar phage in heat map analysis) and phiEa21-4 (type phage) (Figure 5). As aforementioned, the functional ORFs of ΦFifi106, vB_EamM-M7, and phiEa21-4 were categorized into four groups. All functional ORFs of ΦFifi106 displaying nucleotide identity (80–100%) with vB_EamM-M7 and phiEa21-4 were indicated as gray shading. Thus, ΦFifi106 was genetically confirmed and classified as a novel member of the genus *Kolesnikvirus* of the subfamily *Ounavirinae*.

### 3.4. Host Specificity and Stability of ΦFifi106

The host range of ΦFifi106 was investigated against thirty-one phytobacteria (Table 1). ΦFifi106 was able to infect all tested *E. amylovora* and *E. pyrifoliae* without any cross-infectivity with other phytobacteria including *P. carotovorum*, *X. arboricola*, and *X. campestris*. The lytic activity of ΦFifi106 was stable at a temperature range of 4 °C to 50 °C and its lytic activity was almost lost at 60 °C (Figure 6A). In addition, its pH stability was well sustained at a pH range of 4 to 10 (Figure 6B). Although its lytic activity under UV irradiation was significantly decreased under a 3 h exposure (*p* < 0.05) (Figure 6C), the overall reduction of phage concentration was only ~0.8 log PFU/mL after a 6 h exposure.

### 3.5. One-Step Growth Curve of ΦFifi106

Prior to the one-step growth curve analysis, the adsorption time of ΦFifi106 was determined to be 20 min (Appendix A). In Figure 7, the phage concentration was maintained at ~4 log PFU/mL for 20 min, after which it was increased significantly to ~7 log PFU/mL until reaching 35 min, finally reaching a plateau (*p* < 0.05). Thus, the latent period and burst size of the phage were determined to be 20 min and 310 ± 30 PFU/infected cell, respectively.

### 3.6. Time–Killing Curves of ΦFifi106 with Different MOIs against E. amylovora

The lytic activity of ΦFifi106 was compared at different MOIs (Figure 8). Without the phage treatment, the bacterial growth was started after 3 h, followed by a further significant growth from 5 to 16 h (*p* < 0.05). Contrary to the control, the bacterial growth was significantly sustained at MOIs of 0.1, 10, and 1000, except for an MOI of 0.001 (*p* < 0.05). Thus, ΦFifi106 controlled the host almost completely and efficiently at MOIs of over 0.1 during a period of 16 h.

### 3.7. In Vivo Evaluation of ΦFifi106 Efficacy for the Control of Fire Blight in M9 Apple Plants

The efficacy of ΦFifi106 on the development of fire blight was investigated in twenty-seven three-month-old M9 apple plants per group. Prior to evaluating the efficacy of ΦFifi106 to control fire blight, the phytotoxicities of ΦFifi106, AgriPhage-FireBlight, and Bramycin were evaluated on the M9 apple plants (Appendix A). None of these treatments caused phytotoxicity in the leaves and stems for two weeks. As shown in Figure 9A,B, the control treated with sterilized water exhibited a disease incidence of 100% with disease severity of levels 1 (11.1%), 2 (40.7%), 3 (18.5%), and 4 (29.6%). Fire blight started from the leaves and expanded to the stems depending on the disease severity in all apple plants (Figure 9C). Conversely, pretreatment of ΦFifi106 significantly reduced disease incidence (37.0%) as well as average disease severity (0.4) (*p* < 0.001). Although few apple plants developed into the level of slight yellow discoloration and/or tissue necrosis, two-thirds of the phage-treated apple plants were found to be in a healthy state. Compared to commercial products, AgriPhage-FireBlight and Bramycin, the disease incidences of both were two times greater than that of our phage. In addition, the average disease severities of AgriPhage-FireBlight (1.4) and Bramycin (1.7) were significantly greater than that of ΦFifi106 (*p* < 0.001), which was confirmed by the systematic development of fire blight (Figure 9C).

## 4. Discussion

Synthetic chemical pesticides, copper-based pesticides, and antibiotics, are commonly employed to prevent phytobacterial infection despite the limitations of the emergence of phytobacterial resistance and environmental issues [5,36,37]. Lytic phages have gained attention as promising alternatives owing to their effectiveness and eco-friendliness [38,39]. Herein, ΦFifi106, against an indigenous *E. amylovora* YKB 14808, was propagated and purified with a final concentration of ~10^11^ PFU/mL, owing to the excellency of lytic activity [7]. To the best of our knowledge, seventy-six tailed *E*. *amylovora*–specific phages have been reported, including fifty-seven myophages (long, rigid, and contractile tail), three siphophages (long, flexible, and noncontractile tail), and nineteen podophages (short and noncontractile tail) [12,12]. The TEM image (Figure 1) revealed that the ΦFifi106 should be classified as a myophage due to its rigid and contractile tail. Compared to other *E*. *amylovora*–specific myophages (Appendix A), the overall head length (79.8 ± 4.3 nm) of ΦFifi106 was much shorter than those of vB_EamM_RisingSun [32], vB_EamM_Joad [32], and pEa_SNUABM_48 [40]. Furthermore, the tail length (114.1 ± 5.2 nm) of ΦFifi106 was shorter than those of vB_EamM_RisingSun [32], vB_EamM_Joad [32], and pEa_SNUABM_31 [40], except for phiEa21-4 [33]. Overall, our phage exhibited relatively shorter head and tail lengths than previously reported *E*. *amylovora*–specific myophages.

The genomic analysis (Figure 2) found that ΦFifi106 consisted of 84,405 bp with a GC content of 43.4%. Sixty-seven hypothetical ORFs, forty-seven functional ORFs, and twenty-six tRNA were predicted. In fact, the possession of tape measure protein and tail sheath protein in ΦFifi106 (Figure 2) characteristics implied the category of myophages [41]. In addition, the absence of its antibiotic resistance, virulence, and lysogenic encoding genes was confirmed from the previously reported databases, thus presenting its safety for the agricultural application. The phylogenetic (Figure 3) and heat map (Figure 4) analyses revealed that ΦFifi106 shared >87% ANI values with the six closest phages (vB_Eam_M-M7, Hena2, phiEa104, phiEa21-4, SunLiRen, and Roscha 1). Based on the threshold ANI value for the classification of the genus (>70%) and species (95%) [28,42], ΦFifi106 belongs to the same genus with the six closest phages and the same species with vB_EamM-M7 and Hena1. As shown in Figure 5, the small amount of similarities (<80%) of our phage with phiEa21-4 (type phage) were found in the host lysis–related protein (endolysin), structure-related protein (tail fiber), and nucleotide metabolism–related proteins (dihydrofolate reductase and deoxynucleoside-5′-monophosphate kinase). Even with vB_EamM-M7 (the most similar phage), ΦFifi106 showed genetic differences in the structure-related protein (tail fiber; 91.9% similarity) and nucleotide metabolism-related proteins (deoxynucleoside-5′-monophosphate kinase; 97.6% similarity). Thus, the novelty of ΦFifi106 was confirmed as a member of the genus *Kolesnikvirus* of the subfamily *Ounavirinae*.

Host specificity of the phage is an important factor to provide successful control over the host and other phytobacteria. Our phage was specific to ten indigenous *E. amylovora* strains and eleven *E. pyrifoliae* strains without cross-infectivity (Table 1). In agreement with other studies [10,14,43], our phage was found to be specific to both *E. amylovora* and *E. pyrifoliae*, which addressed the potential suitability of ΦFifi106 for controlling two representative *Erwinia* strains in Korea.

Diverse and unfavorable environmental conditions can substantially influence the stability of phages [44]. Temperature and pH play a pivotal role in a phage’s attachment, penetration, and amplification inside the host [45]. In addition, UV irradiation can directly affect phages’ replications, owing to an alteration of the phage structure and reduction of infectivity [46,47]. The broad temperature (4–50 °C; Figure 6A) and pH (4–10; Figure 6B) stabilities ensured the stable performance of the phage. According to the Rural Development Administration’s report in Korea, 85.6% of total fire blight outbreaks occurred during May and July with an average temperature range of 11.5–31.5 °C [48]. In addition, the pH condition to be met for the phage was reported in the range of ~5.0–~8.0 [49,50]. Furthermore, the slight loss of the phage concentration over the course of UV exposure (Figure 6C) can be compensated for by applying it with a higher phage concentration or protective formulation (skim milk and sucrose) [51,52]. Thus, the robust lytic stability of ΦFifi106 was proven, as established under various environmental conditions.

The effectiveness of a phage is determined by its fecundity (i.e., adsorption time, latent time, and burst size), MOI, specificity, and stability [53]. As shown in the one-step growth curve (Figure 7), ΦFifi106 exhibited the shortest latent period (20 min) and the largest burst size (310 ± 30 PFU/infected cell), when compared to other reported *E. amylovora*–specific myophages (Appendix A), including pEa_SNUABM_12 (40 min and 18 PFU/infected cell) [10], pEa_SNUABM_47 (40 min and 20 PFU/ infected cell) [10], pEa_SNUABM_50 (40 min and 16 PFU/infected cell) [10], and vB_EamM_Deimos-Minion (180–240 min and 5 PFU/infected cell) [15]. The short latent period with the large burst size will be helpful for accelerating the lysis time and efficiency [13,54]. In addition, the comparison of the lytic activity at different MOIs (Figure 8) revealed that ΦFifi106 significantly suppressed the growth of *E. amylovora* at MOIs of over 0.1 for 16 h after the phage treatment. Compared to phiPccP-1 (6 h, MOI of 0.1) [55] and As-gz (2.5 h, MOI of 10) [56], ΦFifi106 efficiently inhibited *E. amylovora* with a low MOI and relatively long inhibition time.

Although a novel, lytic, and efficient phage demonstrates great potential as an alternative through in vitro characterization, its practical capability is not always persistent when employed in the field [1]. Pretreatment of ΦFifi106 on M9 apple plants most effectively and significantly reduced disease incidence (37.0%) as well as disease severity (0.4) (*p* < 0.001), even compared to AgriPhage-FireBlight (disease incidence of 77.8% and severity of 1.4) and Bramycin (disease incidence of 92.6% and severity of 1.7) (Figure 9). As previous studies employed different treatments and MOIs (Table 2), the disease severity was adjusted with our scale (0 to 4) for direct comparisons. ΦFifi106 demonstrated superior control for the prevention of fire blight with the MOI of 1000, except for ΦEaH5K on the apple plant [16]. Although the treatment of ΦEaH5K was 1,000,000 and 10,000 times greater MOIs, their efficacies were slightly greater than that of ΦFifi106. Interestingly, our phage was more effective than phage cocktails of PEar1, PEar2, PEar4, and PEar6 [18] and ΦH2A, ΦEaH5K, and ΦH7B [11]. More importantly, a single treatment of ΦFifi106 reduced the disease incidence more effectively than a mixture of each phage, ΦEa2345-6 or ΦEa2345-19, and *Pantoea agglomerans* as the phage carrier [17]. Comparatively, ΦFifi106 effectively inhibited *E. amylovora* infection at a relatively low MOI of 1000, highlighting its significant potential for controlling the development of *E. amylovora* infection in apple plants.

## 5. Conclusions

In this study, ΦFifi106 was purified, characterized, and evaluated for controlling fire blight. ΦFifi106, a novel member of the genus *Kolesnikvirus*, showed infectivity to both indigenous *E. amylovora* and *E. pyrifoliae*. ΦFifi106 exhibited excellent lytic activity under broad temperatures and pHs, as well as the exposure to UV irradiation. Moreover, the phage had a comparatively short latent period and large burst size, highlighting its potential use as an alternative. ΦFifi106 efficiently controlled *E. amylovora* at an MOI of 0.1. ΦFifi106 significantly reduced the disease incidence and disease severity of fire blight in M9 apple plants, which was superior to AgriPhage-FireBlight and Bramycin. This study has proven that ΦFifi106 was a novel, safe, efficient, and effective alternative to control fire blight in apple plants. Further study will be performed to expand the employment of ΦFifi106 in the field.

## Figures and Tables

**Figure 1 biology-12-01060-f001:**
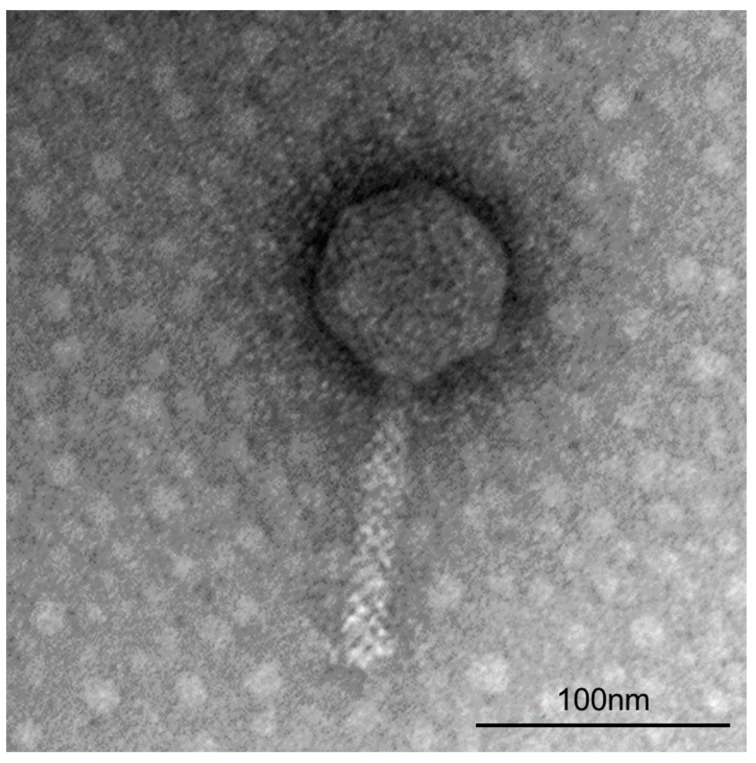
Transmission electron micrograph of ΦFifi106 stained negatively with 2% uranyl acetate at 160,000× magnification.

**Figure 2 biology-12-01060-f002:**
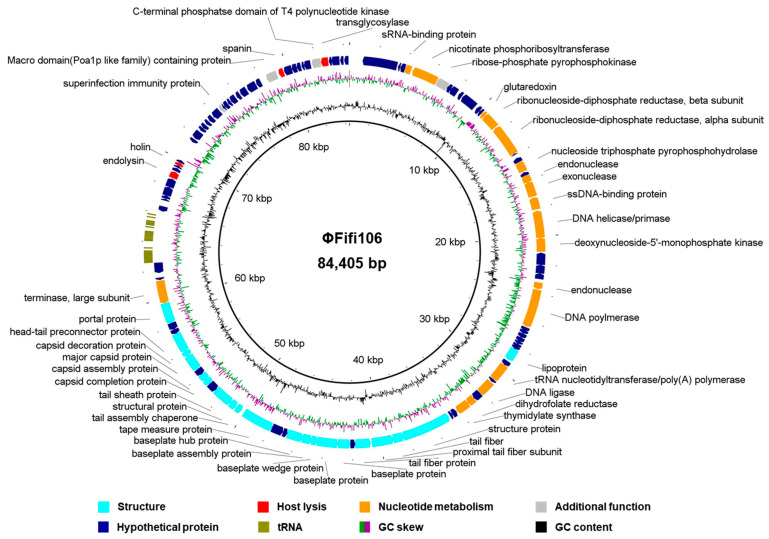
Genome map of ΦFifi106. The predicted ORFs (arrows) and direction of transcription (arrowheads) are indicated. The ORFs were functionally categorized and colored as phage structure (aqua arrow), host lysis (red arrow), nucleotide metabolism (orange arrow), additional functions (gray arrow), hypothetical proteins (navy arrow), and tRNA (olive arrow).

**Figure 3 biology-12-01060-f003:**
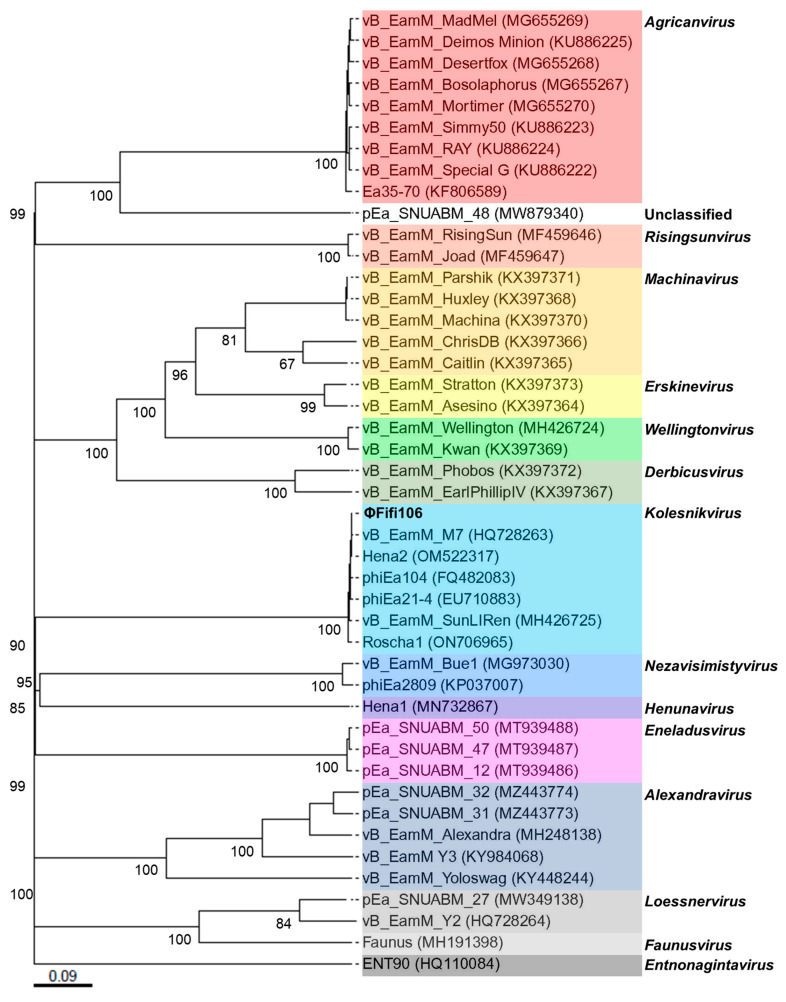
Phylogenetic analysis of ΦFifi106 and forty-four *E. amylovora*–specific myophages. The numbers above the branches are GBDP pseudo-bootstrap support values from 100 replications. The branch lengths of the resulting VICTOR trees are scaled in terms of the respective distance formula used. The genera were listed on the right and indicated using respective colors.

**Figure 4 biology-12-01060-f004:**
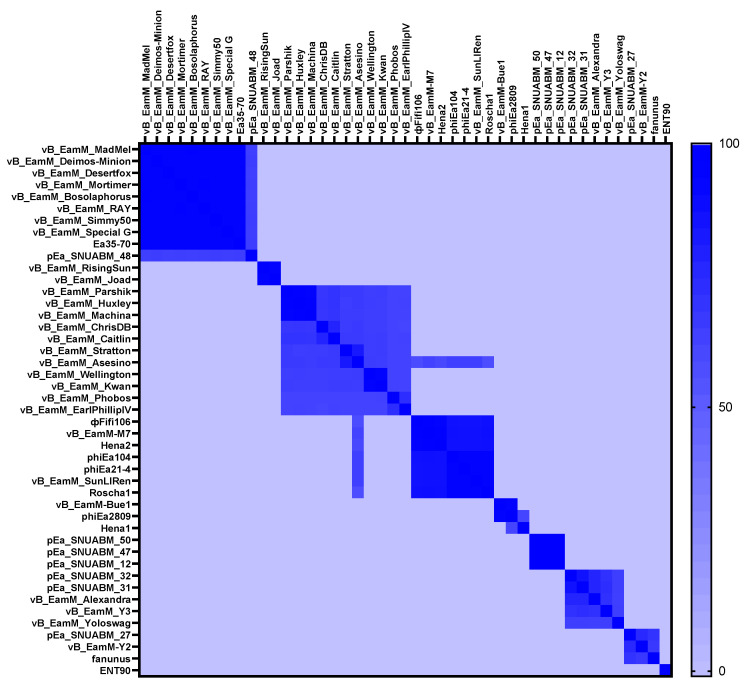
Heat map analysis of average nucleotide identity values of ΦFifi106 and forty-four *E. amylovora*–specific myophages. Variation percent of identity is shown on the color scale.

**Figure 5 biology-12-01060-f005:**
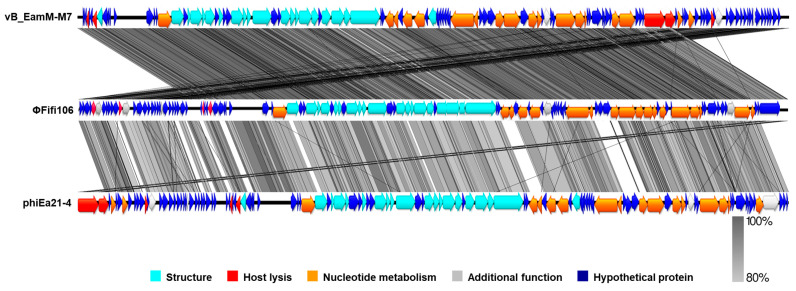
Genomic comparative analysis of ΦFifi106 (middle), vB_EamM_M7 (top), and phiEa21-4 (bottom). Gray shading indicated nucleotide identity between sequences (80–100%).

**Figure 6 biology-12-01060-f006:**
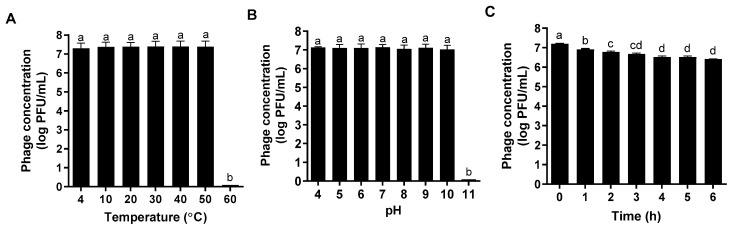
Stability of ΦFifi106 under various (**A**) temperatures, (**B**) pHs, and (**C**) UV irradiation (100 mM/cm^2^). Different letters (a−d) indicate significant differences among treatments (*p* < 0.05).

**Figure 7 biology-12-01060-f007:**
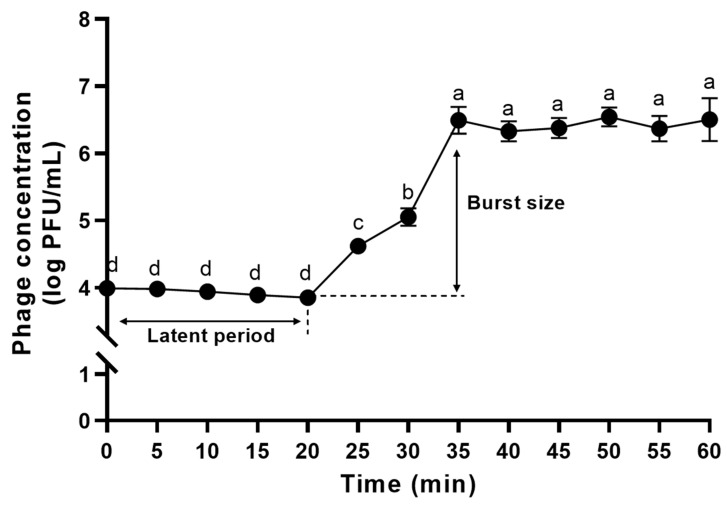
One-step growth curve analysis of ΦFifi106. Different letters (a−d) indicate significant differences among incubation times (*p* < 0.05).

**Figure 8 biology-12-01060-f008:**
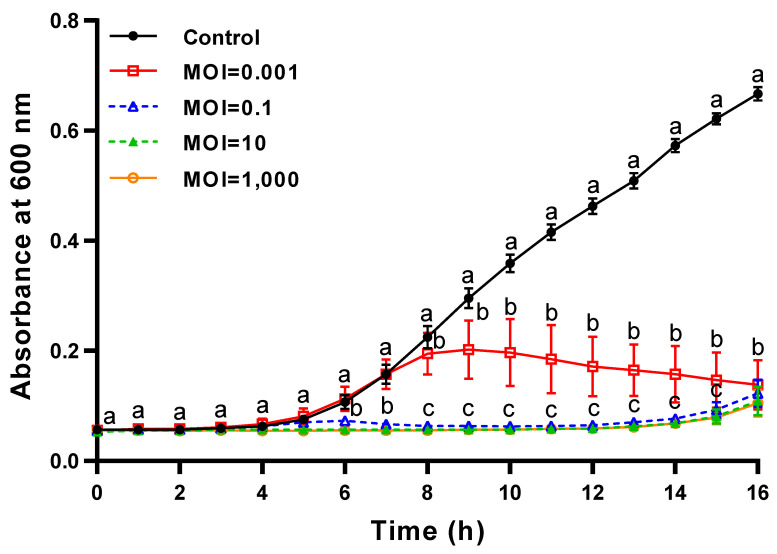
Time–killing curves of ΦFifi106 against *E. amylovora* YKB 14808. *E. amylovora* YKB 14808 was incubated at 28 °C with ΦFifi106-EA at an MOI of 0.001 (open square), 0.1 (open triangle), 10 (closed triangle), and 1000 (open circle), respectively. Different letters (a−c) indicate significant differences among various MOIs within the same incubation time (*p* < 0.05).

**Figure 9 biology-12-01060-f009:**
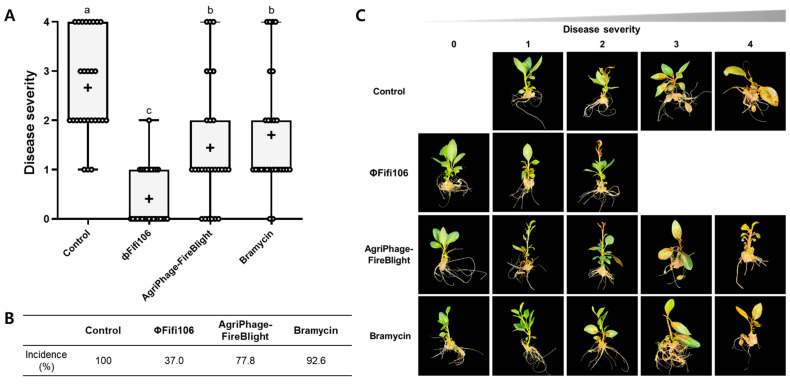
Efficacy of ΦFifi106 for the control of fire blight. The development of (**A**) disease severity and (**B**) disease incidence of M9 apple plants treated with sterilized water (control), ΦFifi106, AgriPhage-FireBlight, and Bramycin, respectively, were evaluated. (**C**) The representative images of M9 apple plants correspond to each level of disease severity after treatment. Different letters (a−c) indicate significant differences among treatments (*p* < 0.001).

**Table 1 biology-12-01060-t001:** Host range of ΦFifi106.

Phytobacteria ^(1)^	Clear Zone Formation ^(2)^	Isolation Source
*Erwinia amylovora* YKB 14808	+	Pear
*E. amylovora* YKB 14748	+	Apple
*E. amylovora* YKB 14750	+	Apple
*E. amylovora* YKB 14758	+	Apple
*E. amylovora* YKB 14776	+	Apple
*E. amylovora* YKB 14787	+	Pear
*E. amylovora* YKB 14814	+	Apple
*E. amylovora* YKB 14818	+	Apple
*E. amylovora* YKB 14820	+	Apple
*E. amylovora* YKB 14822	+	Apple
*E. pyrifoliae* RP0098	+	Pear
*E. pyrifoliae* RP0099	+	Pear
*E. pyrifoliae* RP0100	+	Pear
*E. pyrifoliae* RP0108	+	Pear
*E. pyrifoliae* RP0112	+	Pear
*E. pyrifoliae* RP0113	+	Pear
*E. pyrifoliae* RP0115	+	Pear
*E. pyrifoliae* RP0116	+	Pear
*E. pyrifoliae* KACC 13945	+	NA ^(3)^
*E. pyrifoliae* KACC 13946	+	NA
*E. pyrifoliae* KACC 13947	+	NA
*Pectobacterium carotovorum* KACC 14884	−	NA
*P. carotovorum* KACC 14888	−	NA
*P. carotovorum* KACC 14890	−	Horseradish
*P. carotovorum* KACC 14893	−	NA
*P. carotovorum* KACC 16999	−	Calla
*P. carotovorum* KACC 17004	−	Cabbage
*Xanthomonas arboricola pv. pruni* KACC 18153	−	Apricot
*X. arboricola* pv. *pruni* KACC 18154	−	Apricot
*X. arboricola* pv. *pruni* KACC 18155	−	Apricot
*X. campestris* ATCC 33913 ^(2)^	−	Brussels sprout

^(1)^ YKB and RP were isolated from fire blight-infected apple or pear trees during 2015–2020 in Korea; KACC, Korean Agricultural Culture Collection; ATCC, American Type Culture Collection. ^(2)^ +, clear zone formation; −, no clear zone formation. ^(3)^ NA, not available.

**Table 2 biology-12-01060-t002:** In vivo experiments of *E. amylovora*–specific phages as a pretreatment method (2010 to the present).

Host	Experiment	Phage	MOI	Host (CFU/mL)	Treatment	Time (Day)	Temp. (°C)	Result			Reference
								Scale	Control	Phage	
*E. amylovora* YKB14808	M9 apple plant (*Malus* spp.)	ΦFifi106	10^3^	10^5^	Spray	14	25	Severity (0−4)	2.6	0.4	This study
								Incidence (%)	100	37.0	
*E. amylovora* Ea1	Pear slice (*P. communis* L. Conference)	Cocktail of PEar1, PEar2, PEar4, and PEar6	10^0^	10^6^	Soak	7	− ^(1)^	Severity (%)	58.3	33.8	[18]
			10^1^						58.3	16.6	
			10^2^						58.3	11.7	
*E. amylovora* Ea1/79Sm	Pear slice (*P. communis* L. Jules Guyot Dr.)	Cocktail of ΦH2A, ΦEaH5K, and ΦH7B	10^5^	10^5^	Soak	4	28	Severity (0−6)	3.5	2.1	[8]
	Pear slice (*P. communis* L. Conference)		10^5^						4.2	3.0	
*E. amylovora* Ea1/79Sm	Pear slice (*Pyrus* spp.)	ΦEa1h	10^6^	10^2^	Soak	5	28	Severity (0−3)	2.2	1.3	[19]
		ΦEa100	10^6^						2.2	1.0	
		ΦEa104	10^6^						2.2	0.9	
		ΦEa116	10^6^						2.2	1.0	
*E. amylovora* Ea1/79Sm	Root of apple plant (*M. domestica* B. Pinova)	ΦEaH5K	10^1^	10^5^	Drench	5	−	Severity (0−5)	3.8	2.2	[16]
			10^11^						3.4	2.1	
		ΦEa104	10^1^	10^5^	Drench	5	−	Severity (0−5)	3.8	2.7	
			10^11^						3.4	1.3	
*E. amylovora* Ea1/79Sm	Leaf and stem of apple plant (*M. domestica* B. Pinova)	ΦEaH5K	10^−1^	10^5^	Spray	5	−	Severity (0−5)	4.1	2.6	
			10^9^						4.0	0.9	
		ΦEa104	10^−1^	10^5^	Spray	5	−	Severity (0−5)	4.1	2.6	
			10^9^						4.0	1.5	
*E. amylovora* Ea1/79Sm	Cotyledon of apple plant (*M. domestica* B. Pinova)	ΦEaH5K	10^7^	10^6^	Inoculate	5	−	Severity (0−5)	4.5	1.5	
		ΦEa104	10^7^						4.5	2.5	
*E. amylovora* Ea1337 and *E. amylovora* Ea2345	Flower of B9 apple plant (*M. domestica* 8S6923)	ΦEa2345-6	10^1^	10^8^	Spray	8–20	−	Incidence (%)	27.4	12.4	[17]
		ΦEa2345-19	10^1^						27.4	63.7	

^(1)^ −, not presented.

## Data Availability

The genome sequenced of ΦFifi106 was deposited at GenBank under the accession number OR284297.

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
