# Peer review of "Biological and Genetic Characterizations of a Novel Lytic ΦFifi106 against Indigenous Erwinia amylovora and Evaluation of the Control of Fire Blight in Apple Plants"

_biology, 2023, doi:10.3390/biology12081060_

Round 1

Reviewer 1 Report

The authors purified and characterized the ΦFifi106 phage and evaluated its efficacy for controlling fire blight. I think the study is well conducted however there are few recommendations for the authors in order to improve the manuscript quality.

-The quality of the figures are very bad, mainly figures 2,3,4 and 5, kindly  I encourage the authors to improve this issue.

-Also figures 4 and 5 are not well described, please described the figures along the manuscript.

-Figure 9. Please included in all cases the corresponding photograph records before treatment 

-About the idea in lanes 326-327 “….our phage exhibited relatively shorter head and tail lengths among other reported myophages”….what is the consequence or relevance of this asseveration?

Author Response

Thank you very much for your review. The author carefully revised the manuscript based on the reviewer's comments and marked as red color in manuscript. The response to each reviewer’s comment is shown in red text to facilitate the review process.

Reviewer 2 Report

Choe et al. have presented a well written manuscript that is very pleasant to read. In their manuscript, the authors describe a novel Erwinia amylovora phage fifi106, characterized its biology and potential as a biocontrol agent by means of an elaborate in vivo study.

In the materials and methods, I encourage the authors to include the number of trees used for the in vivo study. Currently, there is no sufficient information on the number of replicates.

I was wondering whether the authors have manually curated the ORFs annotated by Prokka? I would encourage to blastp the protein sequence of the automated annotation to get a higher resolution in the annotation. Furthermore, can the authors include the NCBI accession number of the phage? 

Line 232: I would rephrase heat map analysis and use “ANI analysis”.

Line 237: As phage fifi106 shares 98.8% ANI, I would argue that fifi106 belongs to the same species of phage as vB_EamM-M7  (Turner, D., Kropinski, A. M., & Adriaenssens, E. M. (2021). A Roadmap for Genome-Based Phage Taxonomy. Viruses, 13(3), 506. https://doi.org/10.3390/v13030506). I was wondering whether the authors could perform a VipTree and VIRIDIC analysis as well to compare these results with the ANI?

Line 266: Could the authors provide the adsorption data in supplementary?

Can the authors emphasize on the number of apple saplings they used in each treatment group? 

In terms of the figures, please make sure you upload figures with a higher resolution for publication.

Author Response

(The authors gave the same response as above.)
